# First Case of Paraphimosis as a Severe Complication of Monkeypox

**DOI:** 10.3390/vaccines11010063

**Published:** 2022-12-28

**Authors:** Eugenio Milano, Alessandra Belati, Laura De Santis, Flavio Tanese, Antonio Vavallo, Giuseppe Dachille, Daniela Loconsole, Davide Fiore Bavaro, Francesco Di Gennaro, Maria Chironna, Pasquale Ditonno, Annalisa Saracino

**Affiliations:** 1Clinic of Infectious Diseases, University of Bari, 70124 Bari, Italy; 2Urology, Andrology and Kidney Transplantation Unit, Department of Emergency and Organ Transplantation, University of Bari, 70124 Bari, Italy; 3Department of Interdisciplinary Medicine—Hygiene Section, University of Bari, 70124 Bari, Italy

**Keywords:** monkeypox, paraphimosis, unusual presentation

## Abstract

Since May 2022, the monkeypox (MPX) virus has represented an emerging issue due to outbreaks in non-endemic areas. This report presents the first case of paraphimosis caused by an MPX infection during the outbreak. The patient accessed the emergency department for a sudden onset of swelling of the penis and paraphimosis caused by MPX lesions that brought about stenosis of the foreskin. He therefore underwent a dorsal slit procedure with resolution. No antiviral therapy was required. A multidisciplinary approach should be preferred for the management of MPX, due to the possibility of uncommon and disseminated presentations.

## 1. Introduction

The monkeypox virus (MPXV) is a DNA-virus of the Orthopoxvirus group, the Poxviridae family, closely related to smallpox, causing the monkeypox infection (MPX) [1].

In June 1958, MPXV was identified in Denmark in Cynomolgus monkeys from Singapore [2]. The first human case was documented in 1970 in a 9-month-old baby in the Democratic Republic of Congo. Other cases in humans were later identified in other countries of central-West Africa, thus making MPX endemical [3]. Since 2003, there has been an increase in imported and travel-related cases in countries outside Africa [4].

Since 7 May 2022, multiple cases have been reported worldwide, and on 23 July 2022, the WHO declared MPX a “public emergency of international concern”. To date (17 November 2022), more than 80 thousand cases have been reported, mainly in urban areas, and especially among men who have sex with men (MSM) [5].

Clinically, after 5–21 days of incubation, MPX presents with constitutional symptoms, followed 1–5 days later by rashes and skin lesions, with a cranio-caudal course. Lesions resolve over time with crusts and scabs. Unlike smallpox, various stages of the rash may be observed at the same time (macules, papules, vesicles and pustules). Usually, MPX has a favorable outcome, with a death rate from 1 to 10%, progressively increasing in immunocompromised patients. However, several complications have been reported, which include bacterial superinfections and encephalitis [1].

This case report presents a patient with paraphimosis, a severe complication of MPX infection, which to our knowledge has yet to be reported in the literature.

## 2. Case Description

On 20 July 2022, a 30-year-old Colombian MSM, who had been living in Italy for 6 years, was admitted to the Emergency Department of Policlinico Hospital, Bari, Italy.

His medical history started on 12 July 2022, when the patient returned from Portugal, where he had had condomless sexual intercourse. At that time, he noticed the appearance of skin lesions on the penis, as well as on different parts of the body. Consequently, he consulted a private dermatologist, who performed a swab for MPXV detection by scraping penis vesicles, which was processed through a molecular test at the Laboratory of Molecular Epidemiology and Public Health of the Hygiene Unit at the Policlinico Hospital, Bari, Italy. The screening for Orthopoxvirus was performed using a commercial multiplex RT-PCR kit (Real-Star Orthopoxvirus PCR Kits Altona Diagnostics GmbH), while a second RT-PCR assay specific for MPXV was used to confirm the infection [6]. The result was positive for MPXV with a high viral load (Cq, quantification cycle: 17).

Having tested positive, he was advised to quarantine at home, as he did not report constitutional symptoms. However, due to the sudden onset of penis swelling and the inability to reduce the glans into the foreskin, he went to our emergency department on 20 July, where he tested positive for SARS-CoV-2 and was soon hospitalized in our Infectious Diseases COVID-19 Unit.

Upon clinical examination, no particular signs were assessed, except for laterocervical lymphadenopathy and diffuse lesions. About one hundred small, well-circumscribed, raised, umbilicated, itchy lesions were observed diffused on the genital area, scalp, face, neck, back, and arms (Figure 1). The lesions appeared at various stages of development. The penis (uncircumcised) was the organ that presented the greatest number of lesions, resulting in paraphimosis (Figure 1b). One of these lesions evolved into a large necrotic area, below the coronal sulcus (blue arrow, Figure 2), causing swelling of the glans and retraction of the foreskin, forcing it proximally (red arrow, Figure 2), probably triggering the paraphimosis. The patient complained of mild local pain, although he did not report urinary obstruction, thus making urinary catheterization unnecessary.

The patient did not refer to previous paraphimosis.

A urologic consultation was required, and, after an unsuccessful attempt at manual reduction, surgical intervention was indicated.

Upon admission, a complete blood count showed mild leukocytosis, normal levels of hemoglobin and platelets, creatinine of 1.02 mg/dL, normal levels of electrolytes, C-reactive protein of 31.2 mg/L, no hypertransaminasemia and normal hepatic function.

While waiting for surgery, according to the CDC Guidelines for sexually transmitted infections [7] the following investigations were performed: tests for urethral, rectal and pharyngeal infection; syphilis serology; hepatitis markers. Due to a history of receptive anal intercourse, an anoscopy exam was performed revealing HPV infection, but no associated conditions (i.e., anogenital warts). Screening came back positive for urethral *Chlamydia trachomatis* asymptomatic infection, which was treated with a 1 g dose of azithromycin followed by a further dose of 1 g after 12 h. HPV-59 infection was also found, for which the HPV vaccination was offered to the patient. In addition, an HIV serologic test was performed, with a doubtful reaction for anti-HIV antibodies, so, consequently, a confirmatory test and HIV-RNA test were performed with negative results. For the ongoing risk for HIV acquisition, pre-exposure-prophylaxis for HIV was offered to the patient upon discharge. Moreover, the syphilis test resulted positive, but the patient reported having taken doxycycline about 2 months earlier.

The asymptomatic SARS-CoV-2 infection did not require treatment, since the patient had no risk factors for progression.

Due to a mild exacerbation of penile pain occurrence during his hospitalization, the urologists performed a dorsal slit procedure under local anesthesia on 22 July. After reduction, sutures were placed transversely with rapidly absorbing stitches. Recommendations were given, including compressive medication and avoidance of manual retraction of the foreskin. Figure 3 shows the penis 48 h after the incision.

Because of the unavailability of tecovirimat (TPOXX) in Italy, cidofovir and probenecid were requested upon admission. However, they were not administered because of the patient’s clinical stability, the complete resolution after surgery, and the risk of high renal toxicity from cidofovir.

A test of cure for MPXV was performed on the genital lesions. On 29 July, the test came back positive (Cq: 30) (Figure 4), while on 5 August, the test came back negative. Although it has been documented that healing occurs after all scabs have been removed (in fact, patients who have contracted the monkeypox virus infection are currently placed in respiratory isolation until all scabs from the lesions have fallen off [8]), in our case, the molecular diagnostic test was still positive after the scabs had fallen off. However, re-epithelialization was still in progress.

On 7 August, the patient was discharged, with instructions to quarantine as the SARS-CoV-2 test was still positive.

## 3. Discussion

Monkeypox has always been considered rare, but in recent years there has been a progressive increase in human cases, the majority having a history of travel to countries in Europe or North-America rather than in central and western Africa where the virus is endemic [5].

Moreover, while the endemic form of the disease was classically considered a zoonosis, the main route of transmission in this outbreak is sexual. The group considered at higher risk is MSM reporting condomless and sexual promiscuity [9].

Genital lesions are common in MPX, and in about 1 in 10 patients the infection presents with a single genital lesion, as demonstrated in a large recent study on 528 patients with MPX [10].

To our knowledge, ours is the first documented case of paraphimosis caused by MPX.

Gomez-Garberi et al. recently published a case series of 14 patients with MPX genital lesions with different complications (penile edema, genitourinary tract compromission) and degrees of severity. Two patients required surgery to drain concomitant abscesses and one patient required explorative surgery due to the severity of presentation (sepsis) and the presence of perineal cellulitis at the MRI. For this patient, tecovirimat was requested, but denied by Spanish authorities since the criteria for treatment were not met [11].

The need for the pharmacological treatment of symptomatic cases is still debated. According to CDC-Guidance published on 27 June 2022, our patient should be considered a severe case as he required hospitalization and surgery and consequently, the use of an antiviral therapy was indicated [12].

Tecovirimat is approved by the United States Food and Drug Administration (FDA) for the treatment of smallpox, but data about its efficacy on MPX are limited. Clinical trials showed safety and the CDC allowed a “compassionate use” of TPOXX for Orthopoxvirus infections [12]. Unfortunately, TPOXX was not available in Italy. Conversely, cidofovir and probenecid were requested and available for therapy. Cidofovir has shown effectiveness in in vitro and animal studies, but in vivo data are not available. CDC holds an expanded access protocol that allows for the use of cidofovir for the treatment of Orthopoxvirus outbreaks (including MPXV). Cidofovir is burdened by a high nephrotoxicity, resulting in acute kidney injury, sometimes requiring hemodialysis. Due to the patient’s clinical stability and the absence of new lesion onset for more than 72 h after admission and based on the complete resolution of paraphimosis and local pain with surgery, we preferred to avoid cidofovir.

Brincidofovir may be a safer alternative to cidofovir and has shown efficacy in vitro and in animal study, but it is not yet approved by the FDA [11].

Vaccinia Immune Globulin Intravenous (VIGIV) are allowed by the CDC for the treatment of Orthopoxviruses (including MPXV) in an outbreak, but their effectiveness is debated [12].

## 4. Conclusions

The present global outbreak cases of human monkeypox are currently manifesting predominantly as an STI. Genital lesions are often the only manifestation of the disease, and can assume different macroscopical aspects, presenting as single or multiple lesions, ranging from mild to severely complicated forms, even leading to the need for surgery. A multidisciplinary approach is crucial for the best management of patients with severe genital MPX manifestation.

## Figures and Tables

**Figure 1 vaccines-11-00063-f001:**
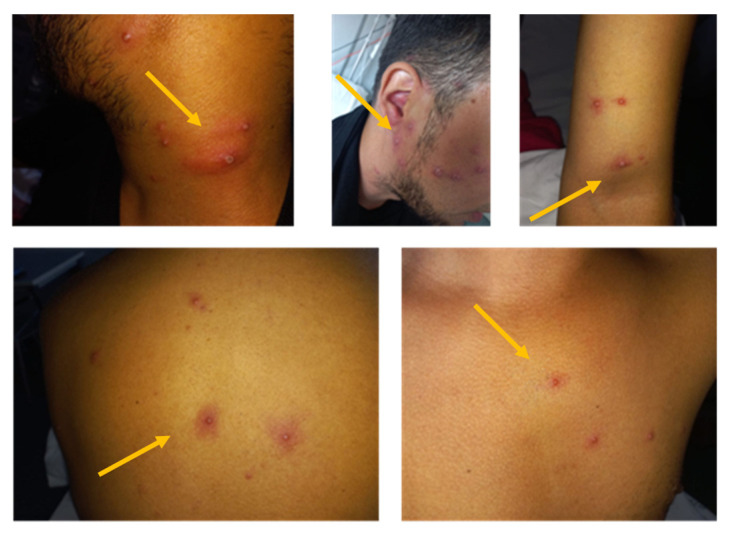
Lesions in different body areas (from left to right: neck, face, arm, back, chest).

**Figure 2 vaccines-11-00063-f002:**
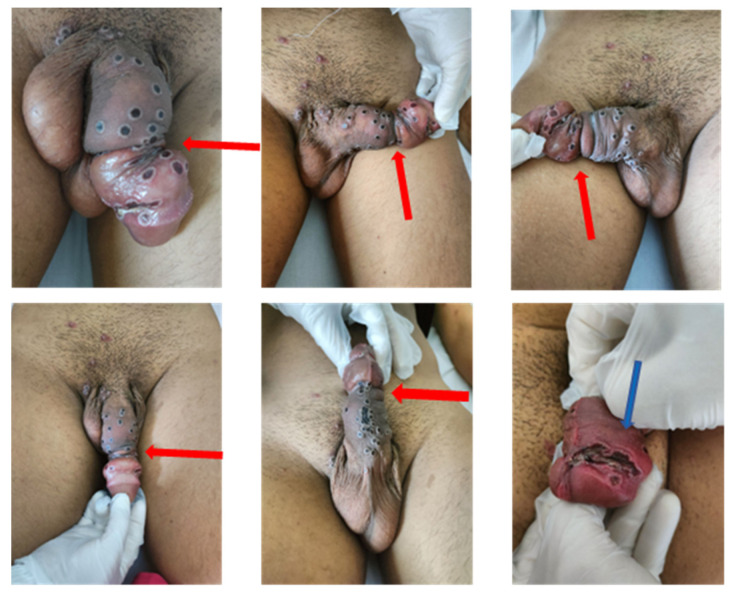
The red arrows indicate, from different perspectives, the penile lesions determining paraphimosis. Blue arrow points to the extensive necrotic area below the coronal sulcus.

**Figure 3 vaccines-11-00063-f003:**
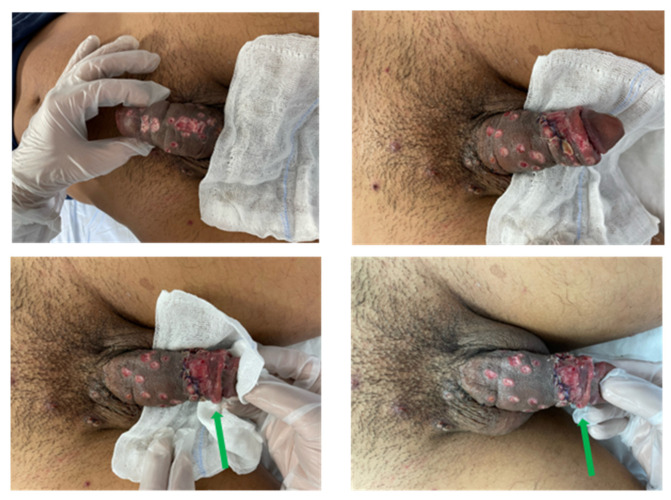
Photos taken 48 h after surgery: previously necrotic lesions are scarring. The largest necrotic area (indicated by the blue arrow in Figure 2) is also healing (green arrow).

**Figure 4 vaccines-11-00063-f004:**
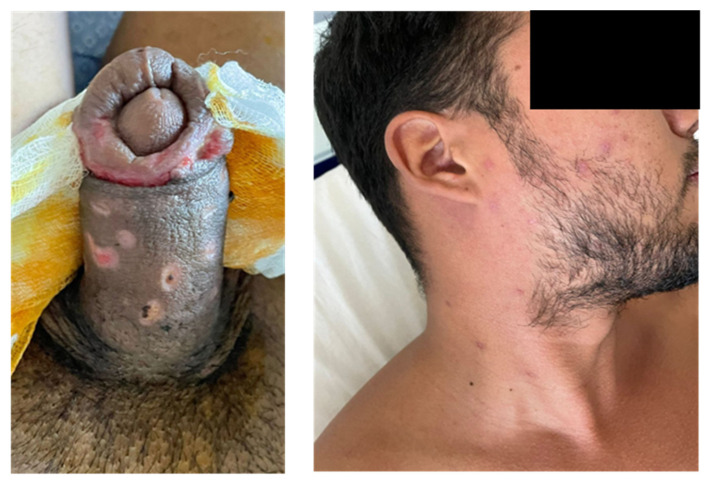
Changes in lesions 9 days following admission. On the left, the penis after surgical intervention: test for MPXV performed on the swab obtained by scraping this area, resulted still positive. On the right, facial lesions on the mend.

## Data Availability

Data can be provided upon reasonable request.

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
