# Peer review of "First Case of Paraphimosis as a Severe Complication of Monkeypox"

_vaccines, 2022, doi:10.3390/vaccines11010063_

Round 1

Reviewer 1 Report

Comments

The study shows a case report focusing on Paraphimosis as a Severe Complication of Monkeypox. The study is attractive however; below mentioned points should be addressed as well as native person needs to revise the English.

Line 13-16: Abstract is short. Abstract should improve with the importance of the disease and brief history of case report following journal format

Line 22-26: Rephrase the sentence: First identified on June 1958, in Denmark, in Cynomolgus monkeys from Singapore [2], the first human case was documented in 1970 in a 9-month-old baby in the Democratic Republic of Congo; subsequently, other human cases were identified in other countries of central-western Africa, where the virus is considered endemic [3], with an increasing increase since 2003 in import and travel-related cases in countries outside Africa [4]

Line 44: where he had condomless sexual intercourses “modify to” intercourse

Line 77-78: Due to history of receptive anal intercourse, HPV infection and associated conditions “add” comma after HPV infection

Line 100-103: Lesions should be mentioned using arrows

Line 104-105: Penis lesions determining paraphimosis “improve” this sentence

Line 107-108: Rephrase the sentence: 48h after incision. Necrotic lesions are scarred. The largest necrotic lesion is also healing (green arrow)

Line 109-111: Rephase the sentence: Evolution of lesions 12 days after admission. In the left, the penis after surgery. On the right, facial lesions on the mend.

Line 161: What is the future research may conduct following your study?

Comment: Check the references carefully following journal format

Comment: What’s about the molecular diagnosis of this case report to differentiate from others infectious diseases?

Author Response

REVIEWER 1

The study shows a case report focusing on Paraphimosis as a Severe Complication of Monkeypox. The study is attractive however; below mentioned points should be addressed as well as native person needs to revise the English.

Thank you for your kind revision.

Line 13-16: Abstract is short. Abstract should improve with the importance of the disease and brief history of case report following journal format

The Abstract has been modified as you suggested. The importance of the disease has been underlined. The case has been detailed.

Line 22-26: Rephrase the sentence: First identified on June 1958, in Denmark, in Cynomolgus monkeys from Singapore [2], the first human case was documented in 1970 in a 9-month-old baby in the Democratic Republic of Congo; subsequently, other human cases were identified in other countries of central-western Africa, where the virus is considered endemic [3], with an increasing increase since 2003 in import and travel-related cases in countries outside Africa [4]

The sentence has been rephrased as follow: “In June 1958, MPXV has been identified in Denmark in Cynomolgus Monkeys from Singapore [2]. The first human case was documented in 1970 in a 9-month-old baby in the Democratic Republic of Congo; other human cases were later identified in other countries of central-western Africa, becoming endemic [3], with an increase in imported and travel-related cases in countries outside Africa, since 2003 [4].

Line 44: where he had condomless sexual intercourses “modify to” intercourse

The word has been modified.

Line 77-78: Due to history of receptive anal intercourse, HPV infection and associated conditions “add” comma after HPV infection

The comma has been added.

Line 100-103: Lesions should be mentioned using arrows

Arrows have been added to the figure.

Line 104-105: Penis lesions determining paraphimosis “improve” this sentence

The sentence has been modified as follow: “The red arrows indicate, from different perspectives, the penile lesions determining paraphimosis. The blue arrow points the extensive necrotic area below the coronal sulcus.”

Line 107-108: Rephrase the sentence: 48h after incision. Necrotic lesions are scarred. The largest necrotic lesion is also healing (green arrow)

The sentence has been rephrased as follow: “Photos taken 48 hours after surgery: previously necrotic lesions are scarring. The largest necrotic area (indicated by the blue arrow in Figure 2) is also healing (green arrow).”

Line 109-111: Rephase the sentence: Evolution of lesions 12 days after admission. In the left, the penis after surgery. On the right, facial lesions on the mend.

The sentence has been rephrased as follow: “Changes in lesions 9 days following admission. In the left, the penis after surgical intervention: test for MPXV performed on the swab obtained by scraping this area, resulted still positive. On the right facial lesions on the mend.” Rechecking the clinical record revealed that 12 days was incorrect. I appreciate you catching this mistake.

Line 161: What is the future research may conduct following your study?

The purpose of our paper is merely to highlight the challenges in diagnosing MPX, particularly when it presents with unusual symptoms. In this regard, we would like to emphasize that, even if it is primely an infectious disease, occasionally a multidisciplinary approach become mandatory, including surgeons.

Comment: Check the references carefully following journal format

References have been checked.

Comment: What’s about the molecular diagnosis of this case report to differentiate from others infectious diseases?

Even while it has been documented that healing occurs after all scabs have peeled (Figure 4), in our case, the molecular diagnostic test was still positive, despite the higher Cq. However, it is currently unknown if healing lesions with still-positive RT-PCR are actually infectious.

Reviewer 2 Report

Some minor phrasing corrections:

Introduction: there were not just ‘several’ cases

- see suggested correction:

“Since May 7th, 2022, multiple cases have been reported worldwide, and on 23rd July 27 2022 the WHO declared the MPX a "public emergency of international concern". To date 28 (17nd Novembre 2022) more than 80 thousand cases have been….”

Discussion: delete this last paragraph:

“Authors should discuss the results and how they can be interpreted from the per- 152 spective of previous studies and of the working hypotheses. The findings and their impli- 153 cations should be discussed in the broadest context possible. Future research directions may also be highlighted.”

Conclusion: rephrase this first sentence as:

“The current global outbreak cases of human Monkeypox are currently manifesting  predominantly as a STI.”

Most ID experts do not regard monkeypox as a true STI

Author Response

REVIEWER 2

Thank you for your comments.

Introduction: there were not just ‘several’ cases

- see suggested correction:

“Since May 7th, 2022, multiple cases have been reported worldwide, and on 23rd July 27 2022 the WHO declared the MPX a "public emergency of international concern". To date 28 (17nd Novembre 2022) more than 80 thousand cases have been….”

The sentence has been rephrased as suggested. 

Discussion: delete this last paragraph:

“Authors should discuss the results and how they can be interpreted from the per- 152 spective of previous studies and of the working hypotheses. The findings and their impli- 153 cations should be discussed in the broadest context possible. Future research directions may also be highlighted.”

This paragraph has been deleted.  

Conclusion: rephrase this first sentence as:

“The current global outbreak cases of human Monkeypox are currently manifesting  predominantly as a STI.”

Most ID experts do not regard monkeypox as a true STI

The sentence has been rephrased as suggested.

Round 2

Reviewer 1 Report

Comment: What’s about the molecular diagnosis of this case report to differentiate from others infectious diseases? answer/improvements should be clear with references

Comment: Native person needs to revise English

Author Response

Thank you for your comments.

What’s about the molecular diagnosis of this case report to differentiate from others infectious diseases? answer/improvements should be clear with references

Thank you for the opportunity to be able to explain the concept better. we have this paragraph, accompanied with bibliographical reference.

“Although it has been documented that healing occurs after all scabs have been removed (in fact, patients who have contracted the monkeypox virus infection are currently placed in respiratory isolation until all scabs from the lesion have fallen off [12]), in our case the molecular diagnostic test was still positive after the scabs had fallen off. However, re-epithelialization was still in progress.”

Native person needs to revise English

We provided a review by a native English speaker.

Round 3

Reviewer 1 Report

Comment: Check the Pictures size, magnification and reset it

Comment: Check the references carefully following journal format